# Reactivity of Small Oxoacids of Sulfur

**DOI:** 10.3390/molecules24152768

**Published:** 2019-07-30

**Authors:** Sergei V. Makarov, Attila K. Horváth, Anna S. Makarova

**Affiliations:** 1State University of Chemistry and Technology, Sheremetevskiy str. 7, 153000 Ivanovo, Russia; 2Department of Inorganic Chemistry, Institute of Chemistry, Faculty of Sciences, University of Pécs, Ifjúság u. 6, H-7624, Hungary; 3G. A. Krestov Institute of Solution Chemistry of the Russian Academy of Sciences, Academicheskaya str., 1, 153045 Ivanovo, Russia

**Keywords:** hydrogen sulfide, sulfoxylic acid, hydrogen thioperoxide, sulfur monoxide

## Abstract

Oxidation of sulfide to sulfate is known to consist of several steps. Key intermediates in this process are the so-called small oxoacids of sulfur (SOS)—sulfenic HSOH (hydrogen thioperoxide, oxadisulfane, or sulfur hydride hydroxide) and sulfoxylic S(OH)_2_ acids. Sulfur monoxide can be considered as a dehydrated form of sulfoxylic acid. Although all of these species play an important role in atmospheric chemistry and in organic synthesis, and are also invoked in biochemical processes, they are quite unstable compounds so much so that their physical and chemical properties are still subject to intense studies. It is well-established that sulfoxylic acid has very strong reducing properties, while sulfenic acid is capable of both oxidizing and reducing various substrates. Here, in this review, the mechanisms of sulfide oxidation as well as data on the structure and reactivity of small sulfur-containing oxoacids, sulfur monoxide, and its precursors are discussed.

## 1. Introduction

Reactive oxygen species, ROS (the most important ROS species are superoxide, hydrogen peroxide, hydroxyl radical, singlet oxygen, hypochlorous acid), occupy a central place in redox biology. They have almost equal importance to reactive nitrogen species, RNS, with nitric oxide (NO) and peroxynitrite ONOO^−^ being the most well-known among these substances. Likewise, in cases of oxygen and nitrogen, they have reactive biologically relevant species. Consequently, sulfur may also form reactive species (RSS) when, for example, a –SH group is oxidized. The concept of RSS was advocated in 2001 [1]. Since then it has become apparent that reactive sulfur species play their own role in cellular function and homeostasis [1,2,3,4,5,6]. In our opinion, the most successful definition of RSS has been proposed by Gruhlke and Slusarenko “RSS are redox-active sulfur-containing molecules that are able, under physiological conditions, to either oxidize or reduce biomolecules” [3]. Reactive sulfur species include nonradical species: Thiols RSH, disulfides RSSR, and polysulfides RS_n_R, sulfenic acids RSOH, thiosulfinates (disulfide-S-monoxides) RS(O)SR, thiosulfonates (disulfide-S-dioxides) RS(O)_2_SR, as well as thiyl radicals RS● [3]. It should be noted that the most well-known groups of RSS are cysteine (Cys) and its derivatives, especially cysteinesulfenic acid (Cys-SOH) [7,8] and cysteine hydropersulfide CysSSH [9], with hydrogen sulfide being one of the most important signal molecules [10], along with disulfides HSSH and polysulfides HS_n_H [4,5]. The parent compounds in these groups are cysteine and hydrogen sulfide, both of which are reducing agents and contain –SH group. But, if for cysteine, it is recognized that oxidative pathways from cysteine leading eventually to cysteinesulfonic acid (Cys-SO_3_H) via successive formation of cysteinesulfenic acid and cysteinesulfinic acid (Cys-SO_2_H) are physiologically important, until 2018 the physiological importance of sulfur species was considered for only the short-lived hydrogen thioperoxide (HSOH) and more stable polysulfides, but not the analogy of cysteinesulfinic acid—sulfoxylic (sulfinic) acid S(OH)_2_. The hydrogen sulfide-disulfide (polysulfides) pathway is considered as the main, if not the exclusive, route of physiologically important transformation of H_2_S [4,5]. It should be noted, however, that sulfoxylic acid is considered as an intermediate of the oxidation of hydrogen sulfide in vitro [11]. In this review we shall discuss the main properties and possible physiological role of the products of oxygenation of hydrogen sulfide—hydrogen thioperoxide and sulfoxylic acid, as well as the product of hypothetical dehydration of S(OH)_2_—sulfur monoxide SO.

Small oxoacids of sulfur (SOS), HSOH, and S(OH)_2_ stand in the middle of so-called sulfur cycle from sulfide to sulfate, between elemental sulfur and sulfite. Though they occupy a central part of this cycle and include compounds containing sulfur in two oxidation states (0, +2), their properties have been scarcely studied. However, recent studies showed [12] that they play a significant role in many important processes, especially in the oxidation of hydrogen sulfide. Sulfoxylic acid (sulfoxylate) is the reactive intermediate of the processes, with participation of industrially important reductants—thiourea dioxide (TDO) and sodium hydroxymethanesulfinate [13]. Sulfur monoxide plays a valuable role in atmospheric chemistry and organic synthesis [14,15]. Recently, we have published a book [16], chapter of a book [17], and a review [18] where, in detail, properties of sulfoxylic acid, its derivatives, and precursors, especially thiourea dioxides, are discussed. From the appearance of these manuscripts, however, some important papers have been published which, in our opinion, deserve the attention of wider audience and will be discussed in detail. Reviews describing properties of hydrogen thioperoxide and sulfur monoxide are absent at all. These are the main reasons why we decided to present a review combining data of reactive sulfur–oxygen species mentioned above. It should, however, be noted that in this review we do not consider dithionites (compounds with sulfur in oxidation state +3), including sodium dithionite Na_2_S_2_O_4_, since the chemistry of these compounds is much more thoroughly known and has been described recently in our review [19] and book [16]. The more recent papers on sodium dithionite focus on only developing and exploiting its well-known application as a strong reducing agent. We shall mention dithionites (dithionous acid) and the product of its monomerization—sulfur dioxide anion radical SO_2_^●−^ just in connection with the other sulfur species.

Among sulfur compounds under review, sulfoxylates are the most well-known and important species. This could be explained by the practical importance of their precursors—sodium hydroxymethanesulfinate (rongalite) HOCH_2_SO_2_Na (note: its other name is sodium formaldehyde sulfoxylate) and thiourea dioxide (NH_2_)_2_CSO_2_. 

## 2. SOS as the Intermediates of Hydrogen Sulfide Oxidation

In principle, the most affordable routes to hydrogen thioperoxide and sulfoxylate could be the oxidation of hydrogen sulfide (sulfides) [11,20] or sulfur [21,22] and reduction (chemical or electrochemical) of dithionite S_2_O_4_^2−^ (sulfur dioxide anion radical SO_2_^●−^) [23,24,25]. The initial step in the one-electron H_2_S oxidation to form HSO^●^ (hydroxysulfinyl) radical and the final step in converting SO_2_ to sulfuric acid have been studied extensively [20]. In contrast, very little is known about oxygenated sulfur species in which sulfur is in the intermediate +1 to +4 oxidation states. The first preparation of hydroxysulfinyl radical in the gas phase has been performed by Frank and coworkers [20]. Reactions of HSO^●^ with oxygen, ozone, NO, and NO_2_ have been studied by Tyndall and Ravishankara [26]. Laboratory measurements and astronomical searches for the HSO^●^ radical indicate that this molecule does not achieve significant abundances either in the gas phase or in the ice mantles of dust grains [27]. Hydroxysulfinyl radical was predicted to undergo exothermic reaction with (^3^Σ) O_2_ to yield SO_2_ [20]. Radical HSO^●^ is a reduced form of the hydroxysulfinyl cation HSO^+^. The latter was prepared by protonation of sulfur dioxide with CH_5_^+^. Collisional neutralization of HSO^+^ by organic molecules yielded hydroxysulfinyl radical. The cation radical of sulfoxylic acid (authors [20] refer to S(OH)_2_ as sulfinic acid) was generated by dissociative ionization of dimethyl sulfate. Collisional neutralization of S(OH)_2_^+^ yielded stable molecules of sulfoxylic acid. The formation of hydroxysulfinyl radical and sulfoxylic acid in the gas phase has been proven by mass spectrometry. 

Betterton and Hoffmann supposed a heterolytic (polar) mechanism of reaction between hydrogen sulfide and peroxides [11]. The primary intermediate of this reaction is HSOH:H_2_S + H_2_O_2_ → HSOH + H_2_O(1)
The next step is accompanied by formation of sulfoxylic acid:HSOH + H_2_O_2_ → S(OH)_2_ + H_2_O(2)
On the other hand, reaction of S(OH)_2_ with HS^−^ could account for the formation of polysulfides: HSOH + HS^−^ → HS_2_^−^ + H_2_O(3)

Earlier, Čermák has shown [24] that sulfoxylic acid can be received by electrochemical reduction of sulfur dioxide anion radical SO_2_^−●^(HSO_2_^●^), which is formed in aqueous solutions of sodium dithionite:HSO_2_^−●^ + e → SO_2_H^−^(4)

The formation of sulfoxylate SO_2_^2−^ in the course of sulfide oxidation in solution has been observed by Vairavamurthy and Zhou [28]. They studied the oxidation of 0.1 M sulfide solutions under conditions of high sulfide-to-oxygen ratios using X-ray absorption near edge structure (XANES) spectroscopy and Fourier transform infrared (FT-IR) spectroscopy. Vairavamurthy and Zhou have noted that lowering the pH of solution to 8.5 causes the 918 cm^−1^ absorption to disappear. According to Tossell [29], only the presence of protonated species (OH)SO^−^ in the original solution (with pH 11.5–12) would explain the presence of the 918 cm^−1^ peak and its disappearance upon acidification. Later, Crabtree and coworkers detected sulfoxylic acid using Fourier transform microwave spectroscopy and double resonance techniques, guided by new high-level CCSD(T) quantum chemical calculations of its molecular structure [30]. In accordance with earlier studies [29,31] their electronic structure calculations have shown that the lowest energy configurations of H_2_SO_2_ in vacuum are rotamers of sulfoxylic acid (denoted S(OH)_2_), where the protons are bound to each of the oxygen atoms. Other isomers of H_2_SO_2_ of potential significance are those termed sulfinic acid, where one proton is bound to the sulfur atom and the other to one of the oxygen atoms (denoted: (HS)O_2_H). The relative stabilities of these isomers in solution are not constrained, nor are the isomers of bisulfoxylate: (HS)O_2_^−^ and (HO)SO^−^ [32]. 

Studies on hydrogen thioperoxide are very sparse. It was detected in an argon matrix during the course of photolysis of ozone and hydrogen sulfide [33], and was generated from an H_2_S/N_2_O mixture in a chemical ionization source [34]. But these synthetic routes are suited to prepare only very small quantities of HSOH. Winnewisser and coworkers have suggested another route to prepare hydrogen thioperoxide: Gas-phase HSOH was synthesized by flash vacuum pyrolysis of di-*tert*-butyl sulfoxide [35] (Scheme 1): 

Investigations of the pyrolysis reaction by mass spectrometry, matrix isolation, and gas phase FT-IR spectroscopy revealed that up to 500 °C di-*tert*-butyl sulfoxide decomposes selectively into *tert*-butylsulfenic acid, (*t*BuSOH), and 2-methylpropene. Transient *tert*-butylsulfenic acid has been characterized by a comprehensive matrix and gas phase vibrational IR study guided by the predicted vibrational spectrum calculated at the density functional theory (DFT) level (B3LYP/6-311+G(2d,p)). At higher temperatures, the intramolecular decomposition of *tert*-butylsulfenic acid, monitored by matrix IR spectroscopy, yields short-lived hydrogen thioperoxide along with 2-methylpropene, but also H_2_O, and most probably sulfur atoms [36].

Recently Kumar and Farmer have found that small oxoacids of sulfur sulfenic (HSOH, hydrogen thioperoxide), sulfoxylic (H_2_SO_2_), and thiosulfoxylic (H_2_S_2_O_2_) acids may be trapped in situ by derivatization with nucleophilic and electrophilic trapping agents [12]. The generated SOS are derivable from reactions with nucleophilic traps such as dimedone and 1-trimethylsiloxycyclohexene, as well as electrophilic traps such as iodoacetamide and mono- or dibromobimane. Kumar and Farmer have compared SOS formation from H_2_S oxidation with a variety of biologically relevant oxidants (hydrogen peroxide, hypochlorous acid, metmyoglobin (Mb), microperoxidase (MP-11), hydroxocobalamin (Cbl)) as well as maleic peroxide. All reactions were done in the ratio of 1 mM H_2_S, 1.2 mM oxidant at buffered condition around pH = 7. Peroxides and HOCl were expected to directly form the SOS by O-atom transfer (see reaction 1).

Authors have assumed that metalloproteins and Cbl initially oxidize hydrogen sulfide by outer sphere mechanism via one-electron mechanism, for example:2Cbl(III) + H_2_S + H_2_O → H_2_SO + 2Cbl(II) + 2H^+^(5)

HOCl was the most selective oxidant at producing HSOH, with H_2_O_2_ yielding 3:2 mixtures of HSOH and HOSOH. All of the metalloprotein oxidants generated measurable SOS, with MP-11 the more selective for formation of HSOH over HOSOH. Several polysulfides SOS (H_2_S_n_O_m_) were also observed in these reactions. The harder oxidants were, the more persulfanes were generated; for example, the ranking of observed efficiency of HS_3_OH formation was peroxide > hypochlorite > MP > MP-11 > Mb > Cbl. 

In opinion of Nagy and coworkers [37], the trapping methodology with alkylating agents applied by Kumar and Farmer [12] does not deny the possibility that trapped HSOH could have been produced from the hydrolysis of polysulfides to be reacted rapidly with another sulfide molecule: HSOH + H_2_S → HSSH + H_2_O(6)
to generate disulfide species in analogy with the reaction of cysteine sulfenic acid CySOH with cysteine [38]: CySOH + CySH → CySSCy + H_2_O.(7)

Theoretical calculations suggest a chemical mechanism that takes advantage of the interaction between sulfur oxides, SO_n_ (*n* = 1, 2, 3) and hydrogen sulfide (*n*H_2_S), resulting in the efficient formation of a S_n+1_ particle. Reaction (8) occurs via low-energy pathways under water or sulfuric acid catalysis [39].
SO_n_ + *n*H_2_S → S_n+1_ + *n*H_2_O(8)

Polysulfanes oxides may arise from initially formed SOS reacting with H_2_S:HOSOH + H_2_S → HSSOH + H_2_O.(9)

It should be noted that nowadays a generally recognized opinion exists in the literature, on the basis of mechanistic grounds, about the possibility that sulfide oxidations most likely proceed via sulfenic acid (hydrogen thioperoxide) intermediate species, just as in the case of cysteine (Cys) [37,40,41,42]. The fate of HSOH can either be reaction with another nucleophile (e.g., Cys or HS^−^) or with another oxidant molecule (depending on the relative reactivities and concentrations of the reactants) [43]. Interestingly, in the scheme of HSOH oxidation, which is a part of hydrogen sulfide oxidation, discussed in the review of Alvarez and coworkers [41], the intermediate stages of the oxidation of hydrogen thioperoxide at larger oxidant concentrations were not indicated (see: Scheme 2), i.e., formation of sulfoxylic or dithionous acids are not considered. 

In the case of hydrogen peroxide, the final products depend on the initial ratio of hydrogen peroxide to H_2_S, and consist mainly of polysulfides, elemental sulfur, and, in the presence of oxidant excess, sulfate [40]. Unfortunately, there are only scarce data on the influence of the ratio of hydrogen sulfide and oxidant concentrations on the mechanism of the H_2_S oxidation. One possible example is the reaction of H_2_S with Cbl(III). As mentioned above, at [H_2_S] ≈ [Cbl(III)] experimental conditions on the intermediate stages of their reaction, the formation of hydrogen thioperoxide and sulfoxylic acid was observed [12]. Toohee has shown [44] that hydrosulfide displaces H_2_O or cyanide from cobalamin or cobinamide (Cbi differs from the aquacobalamin by the absence of dimethylbenzimidazole axial ligand) to give Co(II) complex. Salnikov and coworkers have studied the reaction between Cbl(III) (hydroxocobalamin) and H_2_S at large excess of hydrogen sulfide at pH 1–10 range [45]. The reaction proceeds in three steps: formation of the complex between aquacobalamin and hydrogen sulfide, inner-sphere electron transfer with formation of Cbl(II)-S^●−^ complex, and addition of a second molecule of hydrogen sulfide to the reduced cobalamin (the last step is shown in Scheme 3).

A stable complex between H_2_S and Cbl(III) could be generated, but only in strongly acidic solutions. When the aqueous H_2_S-Cbl(III) solution was exposed to bubbling air at pH 7 to 8, the product of H_2_S-Cbl(III) reaction was converted to sulfitocobalamin at a relatively high yield [45]. These data also show that in the course of hydrogen sulfide-quacobalamin reaction, a radical HS^●^ (S^●−^) is formed since hydrosulfide ion HS^−^ is not very reactive but hydrosulfide radical HS^●^ (S^●−^) is highly reactive towards dioxygen, resulting eventually in the formation of sulfite as a product [46,47]: S^●−^ + O_2_ → SO_2_^●−^(10)
SO_2_^●−^ + O_2_ → SO_2_ + O_2_^●−^(11)

In the same year Salnikov and coworkers showed that the main product of the reaction between cobinamide and hydrogen sulfide at [H_2_S] >> [Cbi] is a complex of cobinamide(II) with the anion radical SSH^●2−^ [48]. Thus, data mentioned above show that the relative concentrations of hydrogen sulfide and oxidant strongly influence the composition of products of their reactions.

Besides relative concentrations of the reagents, the compositions of products of hydrogen sulfide oxidation will, of course, depend on the relative reactivity of H_2_S and oxidant toward HSOH. However, there are no data on the kinetics of these reactions. Carballal and coworkers [49] mentioned the rate coefficient for reaction between HSOH and HS^−^, but indicate in the footnote that this value has been reported for cysteine. Rabai and coworkers [42] reported the rate coefficients for reactions of HSOH with H_2_O_2_ and HS^−^ (0.04 and 100 M^−1^s^−1^, respectively), but these data have been determined from simulations, not from direct experiments. The most promising way to get kinetic data is to accumulate HSOH from some sources in aqueous solution. In principle, as a source, a compound with SO (SOH) fragment may easily be conceivable, for example, thiourea monoxides (by analogy with thiourea dioxide as a source of sulfoxylate, see below [16,18]). 

## 3. Thiourea Oxides as the Sources of SOS

Thiourea oxides are formed in the course of oxidation of thiourea by hydrogen peroxide or peracetic acid [16]. Relative stability of thiourea mon-, di-, and trioxides strongly depends on the structure of thiourea (R^1^R^2^)N(R^3^R^4^)NCS. If at least one substituent R is a hydrogen atom, thioureas form relatively stable dioxides and trioxides. Tetraalkylthioureas, for example, tetramethylthiourea form in aqueous solutions relatively stable monoxides in the course of oxidation by bromine, bromate [50], chlorite [51], and hydrogen peroxide [52]; while formation of tetramethylthiourea dioxide and trioxide has not been observed at all. Indeed, higher oxides were earlier shown to be unavailable by means of oxidation of tetramethylthiourea by peroxides and alternative pathways of synthesis should be employed [53,54]. Tetramethylthiourea trioxide has been received in reaction between 1-chloro-tetramethylformamidinium chloride with silver sulfite [53]. Formation of tetramethylurea and sulfate as the final products of oxidation of tetramethylthiourea [52], together with the absence of dioxide and trioxide, show that cleavage of C-S bond proceeds at the monoxide stage. Therefore, it is reasonable to assume that the primary sulfur-containing product of tetramethylthiourea monoxide decomposition is hydrogen thioperoxide: (CH_3_)_4_N_2_CSO + H_2_O → (CH_3_)_4_N_2_CO + HSOH(12)

Thus, thiourea monoxide can be considered as a source of hydrogen thioperoxide in aqueous solution. 

An interesting mechanistic feature may be highlighted in the case of a reaction between hydrogen sulfide and chlorine dioxide [55]. This reaction can be divided into two separate kinetic stages. The first stage is very fast, the second one proceeds much more slowly. Authors have compared their results with the data on kinetics of reactions between chlorine dioxide and thiosulfate or sulfite [56,57]. It is shown that these reactions start with electron transfer from the substrate to chlorine dioxide and with oxygen transfer from chlorine dioxide to the substrate, simultaneously. Authors [55] have shown, however, that analogous initiation step
HS^−^ + ^●^ClO_2_ ⇌ ^●^HS + ClO_2_^−^(13)
does not proceed. Instead of this, the fast formation of the weak adduct ^●^HSClO_2_^−^ is observed: HS^−^ + ^●^ClO_2_ ⇌ ^●^HSClO_2_^−^(14)
The next step is accompanied by formation of sulfide radical and chlorite: ^●^HSClO_2_^−^ + OH^−^ → ^●^S^−^ + ClO_2_^−^ + H_2_O(15)
A parallel decomposition of ^●^HSClO_2_^−^ produces chlorine radical and sulfoxylate:^●^HSClO_2_^−^ → SO_2_H^−^ + ^●^Cl(16)

Importantly, contrary to reactions of hydrogen sulfide with peroxides, authors do not assume formation of HSOH in the intermediate stages of the process.

In contrast to hydrogen thioperoxide there are some data on reactivity of sulfoxylic acid (sulfoxylate) received from direct experiments. The reason is the existence of convenient sources of sulfoxylate in aqueous solution - thiourea dioxide [58,59]. In alkaline solutions (pH > 10) TDO decomposes with formation of sulfoxylate:(NH_2_)_2_CSO_2_ + OH^−^ → (NH_2_)_2_CO + SO_2_H^−^(17)

The only nitrogen-containing compound of TDO in strongly alkaline aqueous solutions is the redox inert urea, therefore it is possible to study reactivity of sulfoxylate without isolation of solid compounds. Importantly, sulfoxylate is relatively stable in alkaline solutions under anaerobic conditions and it is possible to accumulate it in solutions. Table 1 shows that at pH ≈ 9 sulfoxylate is even more stable than TDO [60]. 

The other source of sulfoxylate in aqueous solutions is sodium hydroxymethanesulfinate (HMS). Contrary to TDO, HMS is unstable in acidic solutions: HOCH_2_SO_2_^−^ ⇌ SO_2_H^−^ + CH_2_O(18)

Sulfoxylic acid is also unstable in acidic solutions, therefore HMS is a less convenient source for accumulation of sulfoxylate than TDO. Anyway, accumulation of sulfoxylate in alkaline solutions of TDO and subsequent formation of hydroxymethanesulfinate in reactions of SO_2_H^−^ with formaldehyde (reverse reaction 18) gave a possibility to determine the pK_1_ of sulfoxylic acid (pK_1_ ≈ 8.0) [60]. 

Using of TDO as a source of sulfoxylate provided an opportunity to receive the first data of its reactivity from direct kinetic experiments. It was shown that sulfoxylate produces sulfur dioxide anion radical in reactions with dioxygen, superoxide, and hydrogen peroxide: SO_2_^2−^ + O_2_ → SO_2_^●−^ + O_2_^●−^(19)
SO_2_^2−^ + O_2_^●−^ → SO_2_^●−^ + O_2_^2−^(20)
SO_2_^2−^ + O_2_^2−^ + 2 H_2_O → SO_2_^●−^ + 3 OH^−^ + OH^●^(21)

All of these reactions are very fast and proceed faster than corresponding reactions of sulfur dioxide anion radical [61]. Indeed, sulfoxylate and its source—thiourea dioxide, is a more powerful reducing agent than dithionite (dimer of sulfur dioxide anion radical). Thus, in contrast to dithionite, in alkaline solutions under anaerobic conditions sulfoxylate reduces methyl viologen cation (MV^+^) to fully reduced form [59]: SO_2_^2−^ + MV^+^ → SO_2_^●−^ + MV^0^(22)

This reaction was the first example of chemical reduction of MV^+^ to MV^0^ in aqueous solutions. The strong reducing properties of sulfoxylate are manifested in its ability to react with nitrite in strongly alkaline media [59]. This reaction was accompanied by the formation of dithionite, i.e., a one-electron oxidation of SO_2_^2−^ to SO_2_^●−^, followed by dimerization of the sulfur dioxide anion radical.
SO_2_^2−^ + NO_2_^−^ → SO_2_^●−^ + NO_2_^2−^(23)
2SO_2_^●−^ ⇌ S_2_O_4_^2−^(24)

The possibility to reduce nitrite in strongly alkaline solutions in the absence of a catalyst is an important advantage of sulfoxylate (or its precursor—TDO). Most of the known reductions of nitrite were investigated in acidic and neutral media (see, for instance, [62]). In basic solutions the reaction of nitrite, for instance, with sodium borohydride, did not take place unless Cu(OH)_2_ was present [63]. It was also shown that sulfoxylate is capable of reducing nitrous and nitric oxides as well [59]. 

In alkaline solutions sulfoxylate does not react with sulfite [58]. It is known, however, that in slightly acidic solutions the addition of sulfite to the solution of sodium hydroxymethanesulfinate leads to the fast formation of dithionite [64] due to reactions (18) and (25): SO_2_H^−^ + HSO_3_^−^ ⇌ S_2_O_4_^2−^ + H_2_O(25)

Reducing properties of sulfoxylate (TDO) also allows the possibility to obtain unusual highly reduced metal complexes. Thus, contrary to dithionite, sulfoxylate gives a possibility to reduce formally Fe(I)(TSPc)^5−^ to Fe(I)(TSPc●)^6−^, where TSPc is tetrasulfophthalocyanine [65]. EPR spectroscopy was employed to demonstrate formation of reduced states of metallophthalocyanines in the course of their reactions with sulfoxylate [66]. At the same time, the catalytic effect of cobalt macroheterocyclic complexes in reduction of nitrite by TDO has been determined [67,68]. Higher reaction rate with sulfoxylate than with dithionite was observed in the studies of reduction of *μ*-nitrido- and *μ*-oxo-bridged iron phthalocyanines [69]. 

The kinetics of individual stages of reductive interaction of hydroxocobalamin with sulfoxylate in aqueous solutions, Co(III) → Co(II) and Co(II) → Co(I)), have been investigated spectrophotometrically. The rate-limiting step in both reactions is the formation of intermediate complexes of sulfoxylate with Cbl(III) and Cbl(II), respectively [70].

As mentioned above, thioureas form stable dioxides only if in nitrogen containing fragments there is at least one hydrogen atom, i.e., thiourea dioxide should be stabilized by hydrogen bonds [71]. The existence of strong hydrogen bonds explain the relatively low solubility of TDO in water (about 2.5 g in 100 mL of water); besides, the presence of networks of hydrogen bonds in the solid TDO leads to the fact that it is not only poorly soluble in water but its dissolution is also a slow process [72,73]. It was shown recently that even in aqueous solutions TDO exists as a mixture of cyclic clusters of (NH_2_)_2_CSO_2_ and single molecules of its isomer aminoiminomethanesulfinic acid (AIMSA) NH_2_NHCSO_2_H [74]. The slow decomposition of clusters and tautomerization to AIMSA explain the dependence of redox reactivity of thiourea dioxide on the age of its solution (time after dissolution) [75,76,77,78]. 

Modification of TDO structure, for example, insertion of aminoacid residue, leads to the formation of compounds with weaker hydrogen bonds and much higher solubility in water. Indeed, reaction of TDO with glycine in a slightly acidic media in the presence of sodium acetate gives a possibility to synthesize salt O_2_SCNH_2_NHCH_2_COONa [72], which can be used for receiving of much more concentrated solutions of sulfoxylate than can be received from TDO.

## 4. Sulfur Monoxide

Sulfur monoxide (SO) would be the hypothetical dehydrated form of S(OH)_2_, but it is not known to undergo hydrolysis to form sulfoxylic acid [79]. Sulfur monoxide and its dimer S_2_O_2_ are of great interest to astrophysics since they are abundant in the Venusian atmosphere [14]. SO is isovalence-electronic to O_2_ with a triplet ground state and features a S-O bond length of 1.481 Å indicating double-bond character [80]. Sulfur monoxide is formed from SO_2_ under microwave discharge and by oxidation of elemental sulfur [80,81]. Both SO and S_2_O_2_ are unstable—disproportionation of SO leads to the formation of S and SO_2_ [80]. Two ^3^SO produce two S_2_O_2_ isomers OSSO or SOSO, which both have a *cis* and a *trans* conformers. *Cis-* and *trans*-OSSO are both energetically favorable. *Cis-* and *trans*-SOSO are close in energy to free ^3^SO + ^3^SO [14]. The formation and photochemical loss of OSSO have been discussed in detail by Kjaergaard and coworkers [14].

For synthetic purposes in organic chemistry, sulfur monoxide can be received in situ, for example, from thiirane oxide [82] or a trisulfide-2-oxide [83]. SO can be trapped by dienes or metal complexes [84,85,86,87]. Thus, heating trisulfide oxide (1) in the presence of dienes (2) results in transfer of sulfur monoxide to form cyclic unsaturated sulfoxides (3) in good to excellent yields, along with recovery of disulfide (4) (Scheme 4) [83]. Lemal and Chao have presented evidence that the triplet SO is formed exclusively as a thermal decomposition product of thiirane oxide. They have also discussed the mechanism of addition of triplet SO to dienes [88]. 

Stephan and coworkers have shown that a frustrated Lewis pair can be used for activation of *N*-sulfinylamine which is a convenient source of SO [15]. Later Cummins and coworkers reported on synthesis and reactivity of a molecular precursor for SO generation, namely 7-sulfinylamino-7-azadibenzonorbornadiene (**5**) (Figure 1) [89]. This compound releases sulfur monoxide at mild temperatures (<100 °C) and allows for SO transfer, in solution, to organic molecules as well as transition metal complexes. Cummins and coworkers have shown that the formation of singlet SO is thermodynamically strongly favorable in the case of (**5**). ^3^Σ^−^ SO was detected by microwave spectroscopy, possibly originating from ^1^Δ SO [89]. The singlet state is shown to be more reactive than the ground state, in analogy with molecular oxygen [90].

## 5. SOS in Reduction of Sulfite

A very interesting biochemical aspect related to sulfur monoxide and/or sulfoxylate is their participation in reduction of sulfite. At the intracellular level, sulfite is a pivotal intermediate during microbial sulfate reduction (MSR). In contrast to chemical hydrogen sulfide oxidation, which proceeds relatively smoothly, the opposite reaction occurs in the sulfur cycle—reduction of sulfite remains inaccessible to synthetic catalysts [91].

Enzymatic reduction of sulfite (SO_3_^2−^) requires the delivery of six electrons and seven protons (Equation (26)).
SO_3_^2−^ + 6 e^−^ + 7 H^+^ → HS^−^ + 3H_2_O(26)

This reaction is driven by sulfite reductases (SiR) and goes to completion. The enzymes performing this reaction may be classified as dissimilatory (dSir) and assimilatory (aSir) ones. Both types contain a cofactor composed of a heme macrocycle (siroheme) and a cubane [4Fe-4S] cluster bridged by a cysteine residue. In recent biochemical models [92,93,94,95,96] reduction of sulfite at the siroheme-[4Fe-4S] catalytic site in dissimilatory sulfite reductase occurs in a stepwise fashion, first producing a bound S^2+^ intermediate, then a bound S^0^ intermediate, before eventually leading to the formation of sulfide. aSir usually converts sulfite directly to hydrogen sulfide without generating intermediates; while dSir produces or reacts with the other sulfur compounds—thiosulfate S_2_O_3_^2−^ or trithionate S_3_O_6_^2−^ [92]. Reactions of enzymatically bound intermediates (S^2+^, S^0^) via nucleophilic attack by residual nonenzymatically bound sulfite species have been hypothesized as pathways for the generation of thiosulfate and trithionate that have been observed in some MSR culture experiments [32,92].

However, whether trithionate and thiosulfate are really necessary intermediates still remains an open question [93]. Santos and coworkers assumed [94] that previous reports of thiosulfate and trithionate production were the result of nonphysiological in vitro conditions, for instance, unrealistically high (bi)sulfite concentrations. 

Even if we assume that thiosulfate and trithionate are forming, the mechanism of their formation is unclear. As we discussed earlier, the primary product of the reaction between sulfoxylate and bisulfite is dithionite S_2_O_4_^2−^ (see reaction 25), but not thiosulfate or trithionate. The latter products can be formed as a result of decomposition of dithionite. 

The other open question—what is S^2+^ species forming during sulfite reduction, sulfur monoxide or sulfoxylic acid (sulfoxylate)? As mentioned above, there are no data on hydration of SO or dehydration of S(OH)_2_, so there is no interconversion between sulfur monoxide and sulfoxylic acid. Some researchers assume that the reduction of sulfite to sulfide is likely to proceed through a siroheme-bound sulfoxylate (SO_2_^2−^) [92,97,98,99] (see Scheme 5). In this case the intermediate represents a complex of Fe^3+^ with a very strong reductant—sulfoxylate, and an inner electron transfer is very likely.

Finally, we have proposed a catalytic cycle for the enzyme. The last steps of the cycle include the formation of Fe-SO adducts, proposed to be a critical step in this cycle [101]. Linkage isomerism is possible in Fe-SO models and Fe^2+^-SO^0^ can shift to its OS form [102,103]. This FeOS intermediate would lead to a possible alternative route. 

## 6. Conclusions

As can be seen from the discussion, oxidation of hydrogen sulfide may lead to formation of a very reactive reductant—sulfoxylic acid. This differs H_2_S from the other very valuable biologically SH-compound—cysteine. Indeed, the analog of cysteine—cysteinesulfinic acid—practically has no reducing properties, for example, it cannot even reduce aquacobalamin [104]. These differences should be considered when comparing the biological roles of hydrogen sulfide and cysteine. The strong reducing properties of sulfoxylate may also be taken into consideration in the chemistry of sulfite reductases.

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
