# Peer review of "Reactivity of Small Oxoacids of Sulfur"

_molecules, 2019, doi:10.3390/molecules24152768_

Round 1
Reviewer 1 Report
The authors have written an interesting and informative review of the energetics of some seemingly simple sulfur-containing species. It is wondeful to see "small molecules" get such attention for their novel chemistry. I
L. 69, The authors but very briefly discuss Na2S2O4. How about briefly discussing Na2S2O3 and perchance Na2S2O5? These both contain S-S bonds.They both are sulfonic acid derivatives and a brief comparison of [S-SO3]-2, [OS-SO2]-2 and [OS-O-SO]-2 would be interesting and informative, as would [O2S-O-SO2]-2 and [O2S-SO3]-2
L. 92, The proton affinity of SO2 is c a. 130 kJ mol-1 more positive than that of CH4. How is protonation of So2 with CH5+ just "mildly exothermic"?,
L. 202, How am I to understand the bonding in (SSH)2-?
L. 231, A nitrogen is missing in the structure of thiourea
L. 276 The doublehheaded arrow conveys resonance which clearly is not meant.
L. 305, A reaction wit nitrous oxide is mentioned -- what about with nitric oxide
L. 312 How is it known whether it is the metal center of phthalocyanine ligand which is reduced?
L. 341, The authors mention SO is a ground state triplet by analogy to O2 and earlier mention singlet O2. Could the singlet-triplet split of O2 and that of SO be mentioned explicitly. Could the suggested geometry of S2O2 be mentioned as well as discussing, however briefly why SO dimerizes but O2 does not.
L. 346,for both the thiirane and trisulfide oxide precursors, Is the SO formed as a spin-allowed, higher energy singlet or lower energy, spin forbidden triplet reaction?
L. 357, a typo, it is ...norbornadiene, not nonbornadiene
A final, general question, could the chemistry of the isomeric SSO and SOS (yes, this as triatomic) be briefly discussed
Author Response
The authors have written an interesting and informative review of the energetics of some seemingly simple sulfur-containing species. It is wondeful to see "small molecules" get such attention for their novel chemistry. I
L. 69, The authors but very briefly discuss Na2S2O4. How about briefly discussing Na2S2O3 and perchance Na2S2O5? These both contain S-S bonds.They both are sulfonic acid derivatives and a brief comparison of [S-SO3]-2, [OS-SO2]-2 and [OS-O-SO]-2 would be interesting and informative, as would [O2S-O-SO2]-2 and [O2S-SO3]-2 The properties of sodium dithionite were recently discussed in detail in our papers and book 16-19. The properties of thiosulfate will be a topic of a special review
L. 92, The proton affinity of SO2 is c a. 130 kJ mol-1 more positive than that of CH4. How is protonation of So2 with CH5+ just "mildly exothermic"?, corrected
L. 202, How am I to understand the bonding in (SSH)2-? This is a product of reaction of coordinated S.- with HS- having S-S bond.
L. 231, A nitrogen is missing in the structure of thiourea corrected
L. 276 The doublehheaded arrow conveys resonance which clearly is not meant. corrected
L. 305, A reaction wit nitrous oxide is mentioned -- what about with nitric oxide added nitric oxide in the text (line 306)
L. 312 How is it known whether it is the metal center of phthalocyanine ligand which is reduced? The formation of highly reduced species has been proved using EPR.
L. 341, The authors mention SO is a ground state triplet by analogy to O2 and earlier mention singlet O2. Could the singlet-triplet split of O2 and that of SO be mentioned explicitly. Could the suggested geometry of S2O2 be mentioned as well as discussing, however briefly why SO dimerizes but O2 does not. Added lines 345-349б 368-369, as well as ref. 90.
L. 346,for both the thiirane and trisulfide oxide precursors, Is the SO formed as a spin-allowed, higher energy singlet or lower energy, spin forbidden triplet reaction? Added lines 354-356 and 366-368 as well as ref. 88.
L. 357, a typo, it is ...norbornadiene, not nonbornadiene corrected
A final, general question, could the chemistry of the isomeric SSO and SOS (yes, this as triatomic) be briefly discussed This will be a topic of the review on sulfur dioxide
Reviewer 2 Report
The authors present a thoroughly written review in the field of the so-called small oxoacids of sulfur. I recommend the manuscript for publication in MOLECULES.
Author Response
The reviewer recommend publication